# Sensitivity to Social Contingency in Adults with High-Functioning Autism during Computer-Mediated Embodied Interaction

**DOI:** 10.3390/bs8020022

**Published:** 2018-02-08

**Authors:** Leonardo Zapata-Fonseca, Tom Froese, Leonhard Schilbach, Kai Vogeley, Bert Timmermans

**Affiliations:** 1Plan de Estudios Combinados en Medicina (MD/PhD), Facultad de Medicina, Universidad Nacional Autónoma de México, 04510 Mexico City, Mexico; 2Centro de Ciencias de la Complejidad, Universidad Nacional Autónoma de México, 04510 Mexico City, Mexico; t.froese@gmail.com; 3Departamento de Ciencias de la Computación, Instituto de Investigaciones en Matemáticas Aplicadas y en Sistemas, Universidad Nacional Autónoma de México, 04510 Mexico City, Mexico; 4Independent Max Planck Research Group for Social Neuroscience, Max Planck Institute of Psychiatry, 80804 Munich, Germany; leonhard_schilbach@psych.mpg.de; 5Department of Psychiatry, Ludwig-Maximilians-Universität, 80539 Munich, Germany; 6Cognitive Neuroscience, Institute of Neuroscience and Medicine (INM-3), Jülich Research Center, 52425 Jülich, Germany; kai.vogeley@uk-koeln.de; 7Department of Psychiatry, University Hospital Cologne, 50937 Cologne, Germany; 8School of Psychology, University of Aberdeen, Aberdeen AB24 3FX, Scotland, UK

**Keywords:** sensorimotor contingencies, intersubjectivity, autism spectrum disorder, embodied interaction, social interaction, virtual reality, human-computer interface

## Abstract

Autism Spectrum Disorder (ASD) can be understood as a social interaction disorder. This makes the emerging “second-person approach” to social cognition a more promising framework for studying ASD than classical approaches focusing on mindreading capacities in detached, observer-based arrangements. According to the second-person approach, embodied, perceptual, and embedded or interactive capabilities are also required for understanding others, and these are hypothesized to be compromised in ASD. We therefore recorded the dynamics of real-time sensorimotor interaction in pairs of control participants and participants with High-Functioning Autism (HFA), using the minimalistic human-computer interface paradigm known as “perceptual crossing” (PC). We investigated whether HFA is associated with impaired detection of social contingency, i.e., a reduced sensitivity to the other’s responsiveness to one’s own behavior. Surprisingly, our analysis reveals that, at least under the conditions of this highly simplified, computer-mediated, embodied form of social interaction, people with HFA perform equally well as controls. This finding supports the increasing use of virtual reality interfaces for helping people with ASD to better compensate for their social disabilities. Further dynamical analyses are necessary for a better understanding of the mechanisms that are leading to the somewhat surprising results here obtained.

## 1. Introduction

Autism Spectrum Disorder (ASD) is a developmental disorder that is clinically characterized by a diminished ability to communicate, reduced social functioning, and a preference for stereotyped movements and behaviors [1]. High-Functioning Autism (HFA) is a condition that is included in ASD, in which intelligence deficits or learning disabilities are absent and the intelligence quotient (IQ) is in the normal range of the population. However, these preserved intellectual capacities are not sufficient for interacting socially and behaving flexibly as expected in everyday situations [2].

Traditionally, ASD has been studied from a detached observer position focusing on the capacities of “Theory of Mind” (ToM) or mentalizing [3,4]. ToM approaches ascertain that for individuals to understand and interact with the social world, they must infer and impute mental states to themselves and to others. In the case of ASD, it has been proposed that the social difficulties these people face can be explained by a lack or impairment of a ToM [3].

In consequence, much research on social cognition and ASD has focused on ToM testing tools, entailing high-order cognitive tasks based on inferences about others’ mental states [5,6]. However, ToM approaches to ASD have difficulties revealing the cause and the core alterations that exist in ASD [7]. For example, it is known that people with ASD employ mentalizing strategies to compensate for their difficulties in understanding others in a more direct or intuitive manner, while still having trouble in dealing appropriately with the demands of real-time social encounters [8,9]. In fact, there is evidence that people with ASD are capable of passing tests related to ToM in up to 15% to 60% [10,11]. Therefore, it seems that some people with ASD do have a ToM but this still does not compensate for their difficulties with social interaction. It therefore seems that problems with social interaction as such, rather than ToM, are at the core of ASD [7,12].

Thus, ASD can be regarded as a social interaction disorder that hinders people’s capacities of co-regulating their own behavior with that of other human beings [13]. Additionally, people with ASD have difficulties in attuning their bodily movements during real-time social interactions, all of which leads to a reduced flexibility in most everyday situations, and a lack of responsiveness to others and to the contingencies presented in social environments [14,15]. This calls for a change in the theories and methods that have been classically used to study ASD.

Fortunately, in the last decade, there has been a growing recognition that a “second-person” approach to social cognition is required to gain a fuller understanding of human sociality [16,17,18]. More recently, researchers have more often been using two-person experimental setups, in which people can interact in real-time. This paradigm contrasts with the traditional “*offline*” study of social interaction that relies on a detached and observational standpoint; instead, it is an interactive and dynamic one (“*online*”), in which participants can feel engaged and researchers can measure the intrinsic time-varying feature of human behavior [19,20]. The second-person approach suggests that social interaction is a crucial part of social cognition, so its study is of the utmost importance for understanding the fact that ASD is characterized not only by impaired higher-order social cognition abilities but also by difficulties in the bodily dynamic attunement during social encounters [21,22].

Peper and colleagues [23] used the term “bodily connectedness” to describe the core property that is impaired in ASD. This concept essentially refers to the social entrainment of bodily movement between two individuals, and its impairment in ASD is beginning to be explored by studies based on coordination dynamics and spontaneous synchronization [21,22,24]. These proposals nicely align with both theoretical and empirical research from embodied, dynamical, and situated cognition perspectives that emphasize the role of social interaction in social cognition [25,26,27,28].

However, it is worth mentioning that so far such “bodily connectedness” studies have focused only on visuospatial perception [29,30,31], thereby seeing the body from a rather external perspective and in terms of the execution of motor behavior. For instance, in automatic visual perspective taking [30], one is still studying whether persons with ASD draw automatic inferences in ways comparable to controls, without looking at whether active bodily involvement in social interaction, rather than inferential processes, reveals differences or similarities with a control population. In doing so, another crucial aspect of the body, i.e., its lived and living quality, is being overlooked. According to the phenomenological tradition, the body is not just a moving object that occupies a space, but also the living organism through which we access the world and are embedded into it [32]. Moreover, we experience this living body from the inside as our lived body, and it affords us with the feeling of being touched, which is arguably one of the most basic forms of becoming aware of the presence of other subjects [33]. Accordingly, phenomenological approaches to psychopathology regard ASD also as a disorder of *intersubjectivity* pointing to a disruption of empathy as an underlying cause for the impairment in making sense of other people [14,34]. Other people appear perceptually opaque, and this lack of direct social perception motivates the employment of compensatory cognitive strategies that, however, fail in addressing the underlying deficits [8].

Therefore, it is important to extend the bodily connectedness proposal so as to include experimental designs capable of accounting for the interoceptive experience of the lived body, for instance through the sense of touch or of being moved, as explored for example in dance therapies [35,36]. Given that embodied intersubjectivity is crucial for better understanding ASD, “skin-to-skin” interactions [37] must also be considered for complementing experiments on gaze responsiveness and movement coordination, both of which allow only face-to-face interactions.

The aim of the current study is to precisely explore tactile-based interactions and the sensitivity to the responsiveness of others (social contingency detection) in patients with HFA during real-time embodied interaction using a minimalist experimental paradigm known as “perceptual crossing (PC)”, originally developed by Lenay [38]. In the PC paradigm participants are embodied as avatars via a human-computer interface (HCI) that allows them to interact with each other and with distractor objects through a sensorimotor loop. In practical terms, participants with one hand move a cursor in a continuous virtual space, while on the other hand, they receive tactile feedback when their cursor touches an object in that space. They can encounter three different kinds of objects: (i) a static object; (ii) a moving object; and (iii) the other person’s cursor or avatar object (who, at the moment of encounter, hence, also touches/encounters the other). The task is to move around freely in this space, and, whenever one encounters an object, to click *only* in those cases, in which she/he is convinced that it is the other’s cursor. The only way to distinguish between the three virtual objects is to read out their different affordances for interaction, otherwise they resemble each other. The degree to which participants can distinguish between a static and moving object relies on their sensitivity to movement. Crucially, however, their capacity to distinguish between encountering a non-reactive moving object and experiencing a mutual encounter with the other’s cursor requires the capacity to perceive complex action contingencies as they are involved in an ongoing interaction.

Despite its minimalist character, this paradigm and its variations have been shown to be a suitable approach for studying the fundamental involvement of sensorimotor interaction in social abilities, such as the interactional basis of detecting social contingency [39,40], of bodily imitation [41], of social perception and social awareness [25,33], and of joint object perception [42]. A common finding is that the relative differences in stability of interacting with the different kinds of objects serve as a way of guiding higher-level social cognitive capacities in an interactively self-organized manner.

Our main hypothesis was that people with HFA would show difficulties in detecting and reacting adequately to the social contingencies that present themselves during interaction with the other’s avatar, as opposed to control (CTRL) participants. Specifically, we expected (a) HFA participants to show a less marked difference between number of encounters with the CTRL avatar and the moving object (as the number of encounters is mutual, this means that we expect only CTRL participants to have fewer encounters with the moving object); (b) HFA participants to be less proficient in detecting the other, that is, show a smaller number of correct clicks per encounter with the CTRL; (c) HFA participants to be more conservative in terms of clicks.

## 2. Materials and Methods

### 2.1. Participants

We tested 10 dyads of adult participants, each consisting of one healthy CTRL participant and one participant with HFA. Participants within one dyad were matched with respect to sex and age.

The 10 HFA participants (5 male) were between 29 and 54 years of age (M = 42.32, SD = 9.20) and were diagnosed and recruited at the Autism Outpatient Clinic at the Department of Psychiatry of the University Hospital of Cologne in Germany (one participant was diagnosed at the Cologne Autism Therapy Centre).

As part of a systematic assessment, the diagnoses are confirmed by clinical interviews according to ICD-10 criteria by two specialized physicians and were supplemented by extensive neuropsychological assessment. The sample included patients with the diagnosis “Asperger syndrome” according to ICD-10 with an at least average Full-Scale IQ (FSIQ N85, measured using Wechsler Adult Intelligence Scale, WAIS). As such, we use the term HFA to refer to individuals with ASD and a high intellectual level of functioning. None of the HFA participants were taking any psychotropic medications. HFA participants took part in this experiment as part of a series of tests during which they participated in three additional experiments.

The 10 CTRL participants (5 male) were between 30 and 54 years of age (M = 43.00, SD = 9.27) and were recruited by means of poster advertisements around the campus of the University Hospital of Cologne in Germany. They reported no history of psychiatric or neurologic disorders and no current use of any psychoactive medications.

Intelligence in both groups was assessed by the German multiple-choice verbal IQ test (“*Wortschatztest*”, WST). Known to provide a valid and time-effective estimate of intelligence [43,44,45], the WST has been used in previous studies for matching purposes [9,46,47,48].

Two tests that are routinely administered during clinical assessment of patients with autism, the Autism Quotient (AQ), and the Beck Depression Inventory (BDI II, [49]) were also administered to the CTRL group. Because depression is a common comorbidity in HFA [50,51], participants with depressive traits were not excluded from any of the groups. Importantly, none of the participants had a clinical diagnosis of depression at the moment of carrying out this research.

Table 1 presents the difference in scores on all tests. For one HFA and one CTRL participant tests scores on IQ, AQ and BDI could not be obtained, and for another CTRL participant, an AQ score could not be obtained. They were removed from the analyses in Table 1. Crucially, as mentioned previously, all HFA participants had been clinically diagnosed at an earlier stage.

### 2.2. Apparatus

In the PC paradigm, two blindfolded participants are embodied as avatars and interact with each other and with distractor objects in a continuous (i.e., circular) one-dimensional virtual space. In this experiment, we used two Tactos devices [38,52] each one consisting of a computer mouse that participants move with their dominant hand and a Braille-stimulator on which they put the index finger of the other hand (See Figure 1). At an overlap of their mouse cursor (sensor) with any object within the virtual space, the small plastic pins of the stimulator move up in unison.

The shared virtual space is depicted in Figure 2a. Participants can come across with a fixed and a moving object, as well as with an object being the other’s avatar that is controlled by the interaction partner’s mouse. While a participant’s avatar is overlapping with another object (sensor activation), that participant receives a tactile stimulation. Such perceptual crossings will be referred as encounters.

Crucially, the moving object is “attached” to the other person’s avatar, so that they have identical movement patterns, except for the fact that when one participant encounters the other participant’s moving object (henceforth referred as lure), only the former feels it and not the other person (Figure 2b), while when they encounter the other’s avatar, they both feel stimulation (Figure 2c). All types of objects are equally sized and elicit the same type of stimulation so they can only be distinguished interactively through the sensorimotor loop that emerges out of the interaction itself but not through differential sensations via the tactile stimulation.

### 2.3. Procedure

Both participants were in the same room but the setup was aimed at avoiding that they could in any way see, hear or feel what the other person was doing when they were moving, and when they were clicking. To this end they sat at different tables, set approximately 3 m apart, the tables were separated by a large occlude, they wore blindfolds, and they wore noise-insulating headphones.

After reading the instructions they first went through a habituation phase, in which they had brief experiences with, consecutively, a stationary object, an object moving at a slow constant speed, and an object moving at a fast-constant speed. Once this habituation phase was completed, they could ask any questions. The task for the participants was to click only when they met an object they thought was the other’s avatar. At no point, they were told how to go about this task (in line with Auvray et al. [39]), and no cooperative context (such as provided by Froese et al. [25]) was created or suggested. Instead, they were told, “Move around in the space, and whenever you encounter an object that you think is the other, make a click.” Participants interacted during three 5-min blocks, between which they could take a short break, but could not communicate in any way.

### 2.4. Analysis

#### 2.4.1. Visual Inspection

As a first step, we calculated the frequency distribution of the distances between the two participants’ avatars. This was a general assessment to see whether the participants tended to be far apart or rather in proximity. As a second step, we restricted the computation of the frequency distribution by only including those distance in which a click was registered, regardless of its correctness. This last procedure aimed to contrast the unconditional distribution with the ones conditioned to participants’ clicks (CTRL vs. HFA).

#### 2.4.2. Classification of Encounters and Clicks

The study was set up as a 2 (Group: CTRL vs. HFA) × 3 (ObjectType: Fixed, Lure, Avatar) × 3 (TrialNumber: 1 to 3) mixed design, with repeated measures on the last two factors. As in Auvray et al. [39], the encounters were defined as both the overlap of at least one pixel between the receptor field and any of the three objects, and the simultaneously associated activation of the motor that would convey the haptic feedback.

We focused on the following dependent variables, in line with previous work on perceptual crossing [25,39]:
(1)Frequency of discrete encounters, which reflects interaction dynamics;(2)Frequency of discrete clicks, which reflects conscious identification;(3)Ratios of clicks to encounters according to ObjectType, which essentially indicates the propensity to click on a specific type of encountered object.

#### 2.4.3. Statistical Analysis

All dependent variables were subjected to a linear mixed effects model (in R, using the lme4 package [54]), including fixed effects of ObjectType (Fixed, Lure, and Avatar), Group (CTRL vs. HFA), and TrialNumber (1 to 3), as well as all possible interactions, and a random effect of DyadNumber on the intercept, accounting for the data originating from different dyads, which differed predominantly in the overall amount of movement and numbers of clicks. In contrast to Auvray et al. [39], we analyzed the raw frequencies instead of proportions of encounters and clicks attributed to each ObjectType per individual, since the latter would obscure any inter-Group differences as well. We used an orthogonal Helmert-type contrast on ObjectType and a polynomial contrast on TrialNumber. The former first compared Fixed with Lure+Avatar, giving an estimate of participants’ sensitivity to object movement, and then, crucially, compared Lure with Avatar, giving an estimate of whether reactivity of a stimulus rather than mere animation elicited encounters or clicks.

This linear mixed effects model was applied to the encounters, the clicks, and the propensity of clicking, which is the number of clicks to the number of encounters. Effect sizes for contrasts were calculated as Pearson *r* based on the *t*-values associated with the model coefficients as obtained by the aforementioned contrasts.

## 3. Results

### 3.1. Frequency Distributions as a Function of the Distance between Participants

In Figure 3 three frequency distributions are shown, corresponding to distances between avatars’ positions and to distances conditional to the presence of CTRL and HFA clicks. A Kolmogorov-Smirnov test indicated that the two sets of distances are drawn from a different distribution (*D* = 0.132, *p* < 0.0001). Overall, the CTRL group has more clicks. The crossings between participants correspond with the values around 0; the crossing of controls with the lure of the participants with HFA corresponds with the values around −50, whereas the opposite situation corresponds with values around +50. Caution is advised when interpreting the plot for the two different populations, as this spatial representation does not take into account participants’ click reaction times. As such, participants clicking faster will have a higher frequency of small distances. Conversely, slower participants will have a correct click represented as a larger distance.

### 3.2. Number of Stimulations Per Object Type and Case of Clicks

For encounters, understood as stretches of time series samples in which the tactile feedback was continuously activated. (Figure 4a), we found a main effect of ObjectType, *F*(2, 153) = 37.3, *p* < 0.0001, and a main effect of TrialNumber, *F*(2, 153) = 7.27, *p* = 0.0010; no other main or interaction effects were present. Planned contrast showed a significant large-sized increase from Fixed to Lure + Avatar (movement), *t*(153) = 8.25, *p* < 0.0001, *r* = 0.55, and a significant medium-sized increase from Lure to Avatar (reactivity), *t*(153) = 2.54, *p* = 0.012, *r* = 0.20. Participants on average encountered the fixed object 62.0 [95% CI: 45.1–78.9] times, the lure 86.0 [95% CI: 69.1–102.9] times, and the other’s avatar 96.4 [95% CI: 79.5–113.3] times. As such, the number of encounters depended both on movement and reactivity, but to a larger degree on movement. The main effect of TrialNumber reflected a significant, medium-sized, linear increase of encounters over the course of the experiment, *t*(153) = 3.67, *p* = 0.0003, *r* = 0.28, with encounters in trial 1 at 72.7 [95% CI: 55.8–89.6], in trial 2 at 83.9 [95% CI: 67.0–100.8], and in trial 3 at 87.7 [95% CI: 70.8–104.6]. As this increase did not interact with any other factors and was not central to our hypotheses, the data in Figure 4 have been collapsed across TrialNumber for the following analyses. Effect sizes *r* for unreported main- or interaction effects involving Group did not exceed 0.10 (small effect threshold).

For clicks (Figure 4b), we again found a main effect of ObjectType, *F*(2, 153) = 35.1, *p* < 0.0001, and this time a main effect of Group *F*(2, 153) = 15.5, *p* = 0.0001; no other main or interaction effects were present. For the effect of ObjectType, planned contrast showed a significant medium-sized increase from Fixed to Lure+Avatar (movement), *t*(153) = 4.34, *p* < 0.0001, *r* = 0.33, and a significant large-sized increase from Lure to Avatar (reactivity), *t*(153) = 7.16, *p* < 0.0001, *r* = 0.50. Participants on average clicked on the fixed object 2.9 [95% CI: 1.3–4.5] times, on the lure 3.0 [95% CI: 1.4–4.6] times, and on the other’s avatar 9.8 [95% CI: 8.2–11.4] times. The overlapping confidence intervals show that the clicks, in contrast to encounters, depended essentially on reactivity. The effect of group was medium-sized (*r* = 0.30), showing on average a higher number of clicks by the Control group [6.7; 95% CI: 5.3–8.1] than by the HFA group [3.7; 95% CI: 2.3–5.1], possibly reflecting a more cautious strategy on the part of the latter. Effect sizes *r* for unreported main- or interaction effects involving Group did not exceed 0.10 (small effect threshold).

### 3.3. Ratio of Clicks to Encounters

The ratio, clicks/encounters, essentially indicates the proportion of object encounters resulting in a click (Figure 4c). Here, we again found the main effect of ObjectType, *F*(2, 153) = 23.7, *p* < 0.0001, and again a main effect of Group, *F*(2, 153) = 11.2, *p* = 0.001. For the effect of ObjectType, planned contrast showed a significant medium-sized increase from Lure to Avatar (reactivity), *t*(153) = 6.61, *p* < 0.0001, *r* = 0.47. Participants’ propensity to click after an encounter was, on average, as follows: for the fixed object 0.06 [95% CI: 0.03–0.08]; for the lure 0.04 [95% CI: 0.01–0.06]; and for the other’s avatar 0.11 [95% CI: 0.08–0.13]. The overlapping confidence intervals show that this ratio, like clicks, but unlike encounters, depended essentially on reactivity: participants clicked relatively more on the avatar than on any other encountered object. The effect of group was medium-sized (*r* = 0.26), and shows a higher number of clicks/encounter by the CTRL group [0.08; 95% CI: 0.06–0.11] than by the HFA group [0.05%; 95% CI: 0.03–0.08%]. Effect sizes *r* for unreported main or interaction effects involving Group did not exceed 0.10 (small effect threshold).

In summary, both CTRL and participants with HFA were equally able to successfully complete the task. Both clicked significantly more often on the other’s avatar than on either of the other two objects they encountered, thereby actually performing better than some of the other PC studies reported in the literature. The HFA group clicked more conservatively, but we cannot rule out that this is an effect of depressive traits rather than of ASD, given the significantly higher scores in the BDI (see Table 1).

## 4. Discussion

In the present study, we aimed to assess the sensitivity to social contingencies in HFA individuals during minimalist computer-mediated embodied interactions.

As the main result, the current study suggests that in context of dynamic interaction in a minimalist virtual environment such as the PC, individuals with HFA are able to detect an agent that is responsive to their own actions and distinguish it from mere non-reactive movement. Indeed, both CTRL and HFA participants were able to distinguish the other person’s avatar from the fixed object and the lure (non-reactive moving object), which is different from the findings by Auvray et al. [39], but in line with Froese et al. [25].

All participants did have somewhat more encounters with the other person’s avatar as compared to the lure (both much more than with a fixed object), but not to a degree observed by Auvray et al. [39]. Despite this comparable number of encounters with the other’s avatar and the lure, where they excelled was the proportion of clicks in response to the other person’s avatar, both in absolute terms, and relative to the number of encounters. This suggests an ability to detect social contingencies when the environmental complexity is greatly reduced.

It is important to mention that the HFA group clicked considerably less in absolute terms. As such, whereas they displayed a sensitivity to properties specific to the other’s avatar that are comparable with CTRL participants, they have a more conservative response bias, something which was observed with autistic individuals in other tasks [55] and in different clinical groups [56], but which may be related to HFA’s comorbidity of depression that is evident from higher BDI scores [57,58].

However, the difference in response bias might also be reflecting the existence of different strategies between groups for solving the task. Such a difference is also suggested by the comparison of distances between avatars at the time of clicking (see Figure 3). Whereas the overall lower number of clicks is visible here as well, the narrower central peak (distance around 0) with respect to the CTRL group in the absence of proportionally higher detection rates in the main analysis, suggests that the HFA group clicked faster after encountering the other person’s cursor. This may reflect a different decision process.

A comparative analysis of the successive trials did not reveal additional information about the overall performance. The lack of an effect of the trial number might be explained by the long-lasting sessions (5 min), which could already include within the very first trial the learning curve that would be expected in any other behavioral task. This methodological aspect contrasts with the PC carried out by Froese et al. [59], in which 15 trials of 60 s each were performed, allowing them to make an informative diachronic analysis on a trial per trial basis.

The results were, therefore, unexpected in two important respects. First, in contrast to earlier studies using this version of the PC paradigm, both the CTRL and HFA group not only clicked more often on the other’s avatar overall but were also significantly more likely to click in response to an encounter with the other’s avatar compared to the other objects. It is not clear why the participants in CTRL-HFA dyads should be more sensitive to social contingency than participants in CTRL-CTRL dyads. This is a topic for future investigation. However, given the good performance, we encountered a second surprise, namely that this clicking performance is not different between the CTRL and HFA groups. In spite of the absent group effect (except for response bias), we were able to point out some advantages in using the PC for studying HFA. We offer several considerations:

If indeed it is the case that persons with HFA are performing similarly to CTRL in the current task (which isolates the reactivity aspect of social interaction), then it is plausible to suggest that the self-organization of this interactive process and the emergence of *coupled units* from the behavior of both interaction partners scaffold the required social judgments to succeed in the task and that this is also possible for persons with HFA [60,61]. This notion is supported by empirical evidence of joint improvisation patterns in adults with ASD during open-ended interactive tasks [62]. Furthermore, theoretical support of this view is provided by the enactive approach to social cognition, in which social interaction is regarded as constitutive for the former [28].

Additionally, the equivalence in performance might be also related to the human-computer interface of the present paradigm. It has been shown that the usage of computer-mediated interactions can support persons with ASD in social engagement [63,64]. The PC paradigm reduces the perceptual and cognitive load that is usually present in daily situations to the basic constituent of social interaction, namely the capacity to detect the specific features of the contingency of a partner with whom I mutually interact. Although if the setup entails dyadic and real-time features, all other additional aspects of social interactions, such as complex and sensory-rich face-to-face interactions do not need to be taken into account and can be fully ignored. Besides that, the participants’ task is very focused and goal-directed, namely to detect different types of contingencies and distinguish those responses caused by another human partner. This maximally reduced setup facilitated the detection of true interaction for persons with HFA, whereas, in contrast, CTRL participants presumably faced a much harder task because they had to judge social interactivity in this abstract manner without additional verbal or nonverbal information of the interaction partner and because HFA individuals presumably provided them with fewer signals of responsiveness that they usually experience in social interactions. Indeed, relatively impoverished stimuli can allow for persons with HFA to overcome limitations associated with complexities of real-life interaction, as has been demonstrated in the fact that action perception is intact in ASD, when they are presented with those actions as point-light displays [65,66].

Crucially, by incorporating the tactile modality, the PC allows participants to interact and recognize the presence of another by means of the most basic sensorial modality that humans have [37,67], and one that has only recently begun to be systematically investigated [33,68]. To our knowledge, this is the first study in which the bodily action of persons with ASD is assessed based on tactile-based interaction with others, rather than on the more common visuospatial interactions, such as eye-gaze-based tasks and the coordination dynamics approaches to social cognition [21,22,23,24]. We speculate that the feeling of being touched is a sufficiently powerful signal for the HFA group to adequately solve the task of identifying the other, and we suggest that more attention should be paid to the domain of tactile interaction in ASD research.

Therefore, the preserved sensitivity to a social contingency that we observed in the HFA group might also indicate an intact mentalizing capacity in the very restricted “social space” of PC. Consequently, it is plausible to propose that a synergetic effect is taking place in the HFA group: (a) instead of having to manage the potentially open-ended goals of real-life social encounters, there is a pre-specified goal of object discrimination and identification; (b) instead of having to manage the potentially open-ended ways of interacting in real-life social encounters, the minimalism of the human-computer interface reduces the scope of interaction to an absolute minimum of computer-mediated binary sensations and horizontal movement; and (c) instead of having to manage the unfolding of real-life social interactions that are always precariously poised to change, develop, or even breakdown altogether, the maintenance of social interaction is favored in the PC paradigm because of the spontaneous entrainment of avatars searching for each other. However, at this point, we cannot confirm whether such an entrainment qualifies as a social or a purely dynamic one.

Importantly, it is also admissible to hypothesize that, despite the similarity in performance between CTRL and HFA participants, other levels of analysis might yield differences between participants in terms of the stochastic sensorimotor patterns (micro-movements) [69,70]. Furthermore, it is crucial to acknowledge that similar judgments (accuracy in clicks) do not equal to similar cognitive performance, in this case, the actual dynamical patterns derived from an interaction with another human being [25,71].

In conclusion, the richness of real-time dyadic and embodied paradigms is acknowledged for getting a better understanding of social interaction and its disturbances. The minimalistic computer-mediated setup used here is particularly promising as it offers open-ended interactions and a dynamic adaptation between dyads that, based on our results on performance, could be of help for coping with difficulties present in persons with different types of social interaction disorders.

The main limitation of the present research is the lack of a comparison between different types of dyads (CTRL-CTRL and HFA-HFA), as well as the reduced sample size. It is possible that higher-order interactions remain elusive, that they are related to how the HFA group may respond differently to the different objects than the CTRL group, or that their way of responding to said objects evolves differently over the course of the experiment. It should be noted, however, that while all reported effect sizes were medium to large, none of the effect sizes involving Group reached the threshold for a small effect; still, the existence of higher-order effects cannot be excluded based on this small sample. What does seem clear is that, even if both groups could still differ to some extent in *how* or *to what degree* they are sensitive to reactive versus non-reactive moving objects and how this extends over time, both groups *are* sensitive to an object’s reactivity to their own actions.

Nonetheless, because of these limitations, our contribution can be considered as a hypothesis generating and feasibility study. Further analyses need to be done to clarify to clarify the contributions from each individual to the whole dynamics of the interaction in the particular case of HFA and to prove whether the “bodily connectedness” proposal stands also for this experimental paradigm. Time-series analysis of dyadic embodied interaction is a compelling approach that has been able to quantify more fine-grained aspects of the interaction, allowing a much better characterization of it [33,71,72]. The simplicity and yet holistic character of the PC has proven not only to be valuable for studying social interaction in a systematic and thorough fashion, but also may yield new and relevant outcomes that could eventually be the basis for the development of bio-behavioral markers in the field of neuropsychiatry, and therapeutic interventions in the setting of social interaction disorders.

## Figures and Tables

**Figure 1 behavsci-08-00022-f001:**
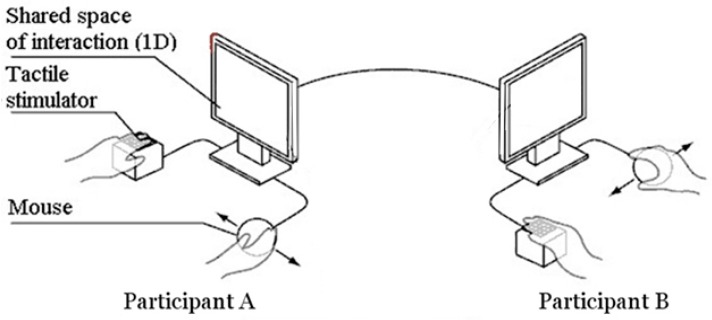
Experimental setup. Participants are physically separated and can only interact with each other through a minimal human-computer interface. They can move a virtual avatar with their mouse cursor with one hand and with the other they can feel an all-or-nothing stimulation whenever their avatar overlaps with an object in the shared virtual space (modified from Lenay and Stewart [53]).

**Figure 2 behavsci-08-00022-f002:**
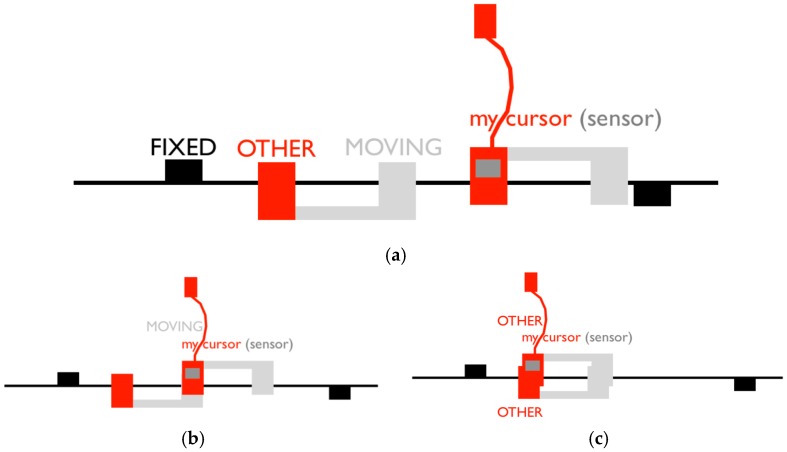
Schematic illustration of the one-dimensional virtual space. (**a**) Participants are embodied as avatars on an invisible line that wraps around after 600 units of space (pixels) in a continuous, circular fashion. Each avatar consists of a sensor (mouse cursor) attached to a body object; (**b**) Unbeknownst to the players, a moving object is connected to each avatar body at a set distance of 50 units. Each participant has her own fixed object located at 150 and 450 pixels, respectively. All objects are 4 pixels wide; (c) A mutual encounter happens when participants are crossing each other, and therefore both simultaneously receive the tactile feedback.

**Figure 3 behavsci-08-00022-f003:**
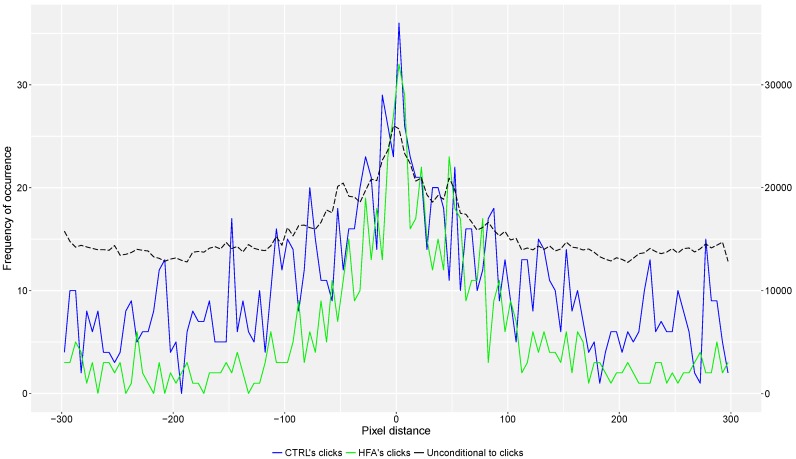
Distances between own and other’s avatar. The frequency distribution of the distances in pixels between the two participants’ avatars is shown. The different colors referred to the overall unconditional to clicks distances, and to those from each group at the time of clicking (4-pixel bins, as objects are 4 pixels wide).

**Figure 4 behavsci-08-00022-f004:**
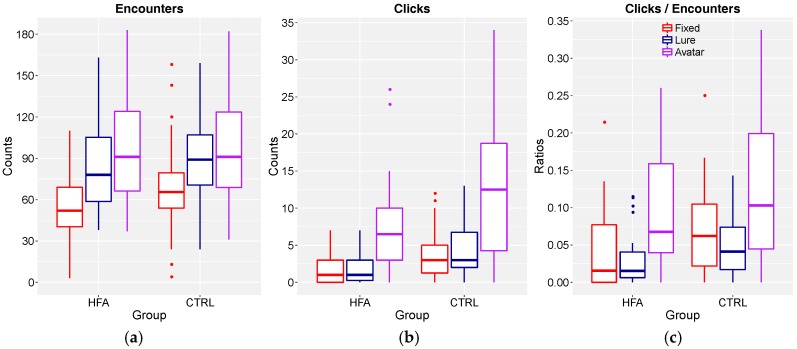
Box-whisker plots of the number of (**a**) encounters, (**b**) clicks, and (**c**) the clicks to encounters ratios. The different colors indicate the object types. Each box represents 30 observations. The middle line inside each box corresponds to the median. The whiskers are in Tukey style (outliers are 1.5 × IQR above or below from the edge of the box). The complete data per dyad/participant and condition, as well as R script, can be found at https://osf.io/h4nkr/.

**Table 1 behavsci-08-00022-t001:** Demographic and neuropsychological data.

	HFA N = 10 (5F/5M)	CTRL N = 10 (5F/5M)	Cohen’s *d*
**Age**	42.32 (9.20)	43.00 (9.27)	*d* = 0.074
**WST**	113.11 (20.13)	109.22 (12.98)	*d* = 0.235
**BDI**	14.11 (9.83)	4.44 (2.65)	*d* = 1.55
**AQ**	40.56 (4.45)	15.63 (3.78)	*d* = 6.06

Mean values and the respective standard deviations; HFA = high-functioning autism group; CTRL = control group; WST = German multiple-choice verbal IQ test (“*Wortschatztest*”); BDI = Beck Depression Inventory (0–13: minimal; 14–19: mild 20–28: moderate; 29–63: severe depression); AQ = Autism Quotient (suggested clinical cutoff 32).

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
