# Peer review of "Sensitivity to Social Contingency in Adults with High-Functioning Autism during Computer-Mediated Embodied Interaction"

_behavsci, 2018, doi:10.3390/bs8020022_

Round 1

Reviewer 1 Report

This study uses a very elegant experimental paradigm to investigate sensitivity to social contingency in adults with high-functioning autism. I very much like the embodied approach to social interaction difficulties, this is a very promising endavour. I do have some comments for improvement.

1.      In the introduction (line 79), the authors describe previous bodily connectedness studies, primarily relying on visuo-spatial perception. It would be helpful to explain this in a bit more detail, in order to better understand how the PC task is different to this or what it is specifically adding. Is the only difference the fact that you include tactile stimuli, or is the task really assessing a different level of attunement?

2.     Line 86 -> the authors describe here ASD as a problem of intersubjectivity and a disruption of empathy. Why is that argument made here and how does this relate to the relevance of using the PC task?

3.     It would be helpful to describe the task in a bit more detail in the introduction, e.g. it only became clear later what was meant for example with distinguishing between the virtual objects (line 108).

4.     The authors describe their main hypothesis rather broadly -> that people with HFA would show difficulties in detecting and reacting adequately to the social contingencies that present themselves during the interaction with the other’s avatar, as compared to control participants. But, what does this mean with regard to their actual study paradigm: would they expect less encounters, less clicks, less accurate clicks. And do these changes reflect a difficulty in detecting or in reacting adequately? Can you differentiate that using this paradigm?

5.     With regard to the task itself, what is the instruction that participants get? And can they click as many times as they want?

6.     The authors use 3 independent variables in their linear mixed effects models, object type, group as well as trial number. They are looking at main effects and possible interactions. However, as you have only 10 participants in each group, I very much doubt whether you have enough statistical power to find a three-way interaction. Why not adding trial number as an additional level in your mixed effects model next to DyadNumber? Further on the authors describe that Trial Number is not really of interest and this would leave only 2 predictors in the model and only one two-way interaction, with fits much better with the statistical power that you have. In addition, it would be helpful to explain how they tested that additional interaction effects (where they all added in one model, or added one by one)?

7.     With regard to the frequency distribution conditional on clicking – and I am not sure whether I am interpreting this right-  it seems to me that the HFA’s not only click less often, but also click more precisely, as they have much less clicks when at a far distance from the other avatar, whereas the CTRL’s show large peaks in clicks even at a large distance. Would the authors agree with that?

8.     The results are really interesting, clearly distinguishing the reactions to the different avatar types, and showing a main effect of group. The authors did not find a significant interaction effect but I really would be interested in seeing the actual effect sizes for the interaction between clicks and Object Type. When looking at the graph, it seems to me that there is a tendency for controls to respond with more clicks to the avatar, but that given the large confidence intervals, this fails to reach significance. It would be interesting to see the numbers here. Do the authors think this could be due to the small sample size? I think it would be fair to add this point to discussion where they are discussing the lack of difference between the two groups (line 358)

9.     The authors also state that the lower level of clicking in the HFA’s may be related to higher levels of depression. As they have the BDI score available, why not controlling for that in the model and see whether the effect is still there?

10.  I would like to see a bit more discussion on what this then actually means. What if indeed HFA’s and CTRLs are not different? The authors now mainly focus on practical and therapeutical benefits of using computer-mediated interactions. But I would be more interested in a more extensive discussion on how this is relevant for the actual understanding of the social deficits in the disorder.   

Author Response

Dear Reviewer, 

Many thanks for the observations. Here our replies:

This study uses a very elegant experimental paradigm to investigate sensitivity to social contingency in adults with high-functioning autism. I very much like the embodied approach to social interaction difficulties, this is a very promising endavour. I do have some comments for improvement.

We thank the reviewer for this positive appraisal

1.      In the introduction (line 79), the authors describe previous bodily connectedness studies, primarily relying on visuo-spatial perception. It would be helpful to explain this in a bit more detail, in order to better understand how the PC task is different to this or what it is specifically adding. Is the only difference the fact that you include tactile stimuli, or is the task really assessing a different level of attunement?

We were referring to studies that look at more inferential processes in both populations; hence we have now added “For instance in automatic visual perspective taking [30], one is still looking at whether persons with ASD draw automatic inferences in ways comparable to controls, without looking at whether active bodily involvement in social interaction, rather than inferential processes, reveals differences or similarities with a control population” (lines 80-83).

2.     Line 86 -> the authors describe here ASD as a problem of intersubjectivity and a disruption of empathy. Why is that argument made here and how does this relate to the relevance of using the PC task?

We agree that it wasn’t clear which idea was introduced in this paragraph. We hope that with the addition mentioned above, it is clear that we contrast more “passive” perceptual qualities related to social cognition, with more embodied, inter-subjective qualities. The PC paradigm, because it allows for dynamic interaction, is one possible technique to address this.

3.     It would be helpful to describe the task in a bit more detail in the introduction, e.g. it only became clear later what was meant for example with distinguishing between the virtual objects (line 108).

Indeed, we overlooked this, it is correct that the paradigm is only described in detail in the method section. To address this in the introduction, we have now added (lines 105-116), “In practical terms, participants with one hand move a cursor in a continuous virtual space, while on the other hand they receive tactile feedback when their cursor touches an object in that space. They can encounter three different kinds of objects: a static object, a moving object, and also the other person’s cursor, or avatar object (who, at the moment of encounter, hence also encounters the other). The task is to move around freely in this space, and, whenever one encounters an object, to click only when one thinks it is the other’s cursor. The only way of distinguishing between the virtual objects is in terms of their different affordances for interaction, because they otherwise resemble each other. Indeed, the degree to which participants can distinguish between a static and moving object is indicative of their sensitivity to movement. Crucially however, their capacity to distinguish between encountering a non-reactive moving object and having a mutual encounter with the other’s cursor is indicative of sensitivity to complex action contingencies involved in live interaction.”

4.     The authors describe their main hypothesis rather broadly -> that people with HFA would show difficulties in detecting and reacting adequately to the social contingencies that present themselves during the interaction with the other’s avatar, as compared to control participants. But, what does this mean with regard to their actual study paradigm: would they expect less encounters, less clicks, less accurate clicks. And do these changes reflect a difficulty in detecting or in reacting adequately? Can you differentiate that using this paradigm?

Indeed, this was rather vague; as such, we now spell out the hypothesis more clearly, “Specifically, we expected (a) HFA participants to show a less marked difference between number of encounters with the CTRL avatar and the moving object (as the number of encounters is mutual, this means that we expect only CTRL participants to have fewer encounters with the moving object); (b) HFA participants to be less proficient in detecting the other, that is, show a smaller number of correct clicks per encounter with the CTRL; (c) HFA participants to be more conservative in terms of clicks” (lines 126-131).

5.     With regard to the task itself, what is the instruction that participants get? And can they click as many times as they want?

They are left entirely free to explore. Their instruction literally was non-specific in terms of strategy: “Move around in the space, and when you encounter an object that you think is the other, click” – indeed, they could click as many times as they wanted. We have added (line 213), “Instead, they were told, “Move around in the space, and whenever you encounter an object that you think is the other, click.”

6.     The authors use 3 independent variables in their linear mixed effects models, object type, group as well as trial number. They are looking at main effects and possible interactions. However, as you have only 10 participants in each group, I very much doubt whether you have enough statistical power to find a three-way interaction. Why not adding trial number as an additional level in your mixed effects model next to DyadNumber? Further on the authors describe that Trial Number is not really of interest and this would leave only 2 predictors in the model and only one two-way interaction, with fits much better with the statistical power that you have. In addition, it would be helpful to explain how they tested that additional interaction effects (where they all added in one model, or added one by one)?

We indeed appreciate the power concerns raised by the reviewer, and the potential to add TrialNr as a random effect; however, we were reluctant to remove the Trial Number factor from the model because first visual inspection of the data suggested there could be a difference in use of strategies over time in both groups (specifically, it looked as if CTRL improved their click/encounter (detection) rate over time, whereas HFA peaked in the second block and then fell back. However, this did not show in the analyses. Precisely because this could have been caused by lack of power (which we now acknowledge in the discussion), we thought it premature to simply exclude it. Also, although many views on this exist, it is recommended that, rather than simplifying the model, the full model is retained; as such, rather than simplifying the model until we observed a loss of explained variance, we opted for keeping the full model. See Barr, D. J., Levy, R., Scheepers, C., and Tily, H. J. (2013). Random effects structure for confirmatory hypothesis testing: Keep it maximal. Journal of Memory and Language, 68(3):255–278 (but see Matuschek, H., Kliegl, R., Vasishth, S., Baayen, H., & Bates, D. (2017). Balancing Type I error and power in linear mixed models. Journal of Memory and Language, 94, 305–315, for a different opinion in the current debate). We also explain briefly why we used a random effect on the intercept (start section 2.4.3): “, which differed predominantly in the overall amount of movement and numbers of clicks” (lines 240-241)

7.     With regard to the frequency distribution conditional on clicking – and I am not sure whether I am interpreting this right-  it seems to me that the HFA’s not only click less often, but also click more precisely, as they have much less clicks when at a far distance from the other avatar, whereas the CTRL’s show large peaks in clicks even at a large distance. Would the authors agree with that?

There are two answers to this. First, the steepness of the central peak is somewhat misleading to the eye, as in fact the difference between the two groups is largely constant over the entire distribution. Careful inspection of the peak shows a similar difference there as on average on the sides (the peaks are noise, which one can smooth out by taking larger bins, but this leads to other interpretation problems). Second, as noted in the text just prior to the picture (line 261), “Caution is advised when interpreting the plot for the two different populations, as this spatial representation does not take into account participants’ click reaction times. As such, participants clicking faster will have a higher frequency of small distances. Conversely, slower participants will have a correct click represented as a larger distance.” — as such, if HFA simply were more focused and clicked faster after a mutual encounter, then the CTRL’s cursor would have been less far away than the HFA were for the slower CTRLs when they clicked.

8.     The results are really interesting, clearly distinguishing the reactions to the different avatar types, and showing a main effect of group. The authors did not find a significant interaction effect but I really would be interested in seeing the actual effect sizes for the interaction between clicks and Object Type. When looking at the graph, it seems to me that there is a tendency for controls to respond with more clicks to the avatar, but that given the large confidence intervals, this fails to reach significance. It would be interesting to see the numbers here. Do the authors think this could be due to the small sample size? I think it would be fair to add this point to discussion where they are discussing the lack of difference between the two groups (line 358)

Indeed, we have added discussion of the impact of sample size to the text: “There are potential issues related to the latter, especially concerning the absence of significant interactions involving Group. It is possible that higher-order interactions remain elusive, that are related to how the HFA group may respond differently to the different objects than the Control group, or that their way of responding to said objects evolves differently over the course of the experiment. It should be noted however, that whereas all reported effect sizes were medium to large, none of the effect sizes involving Group reached the threshold for a small effect; still, the existence of higher-order effects cannot be excluded based on this small sample. What does seem clear is that, even if both groups could still differ to some extent in how they are sensitive to reactive versus non-reactive moving objects and how this extends over time, both groups are sensitive to an object’s reactivity to their own actions” (lines 420-429).  It is not entirely clear what the reviewer means with “interaction between clicks and Object Type” — we assume that they mean between Group and ObjectType. Careful inspection of the plot reveals however that the apparent increment in Avatar clicks for CTRL (with respect to HFA) is simply proportional to the increment in Static/Lure clicks. In fact, whereas the number of clicks on Static/Lure doubles from HFA to CTRL (approx. 2 to 4), this is not quite the case for Avatar clicks (approx. 7 to 12.5). We would like to note that effect sizes for any non-significant effects involving the factor Group were much smaller than the reported ones, and did not exceed the .10 threshold for a small effect (whereas all reported effect sizes are not just small-sized, but rather of medium and large size). We now mention this in the text, at the end of each analysis paragraph (encounters, clicks, and click/encounter). As such, whereas sample size remains an issue, it looks like the differences we report and others related to Group, if the latter exist, would be in a different order of magnitude.

9.     The authors also state that the lower level of clicking in the HFA’s may be related to higher levels of depression. As they have the BDI score available, why not controlling for that in the model and see whether the effect is still there?

In the context of the power issues mentioned by the reviewer, we feared that adding another factor to the model, especially with unequal variances across both groups (apart from being higher in terms of average BDI, the HFA population is also very heterogeneous in terms of how they are affected by depression), would make the results potentially murkier, rather than clearer.

10.  I would like to see a bit more discussion on what this then actually means. What if indeed HFA’s and CTRLs are not different? The authors now mainly focus on practical and therapeutical benefits of using computer-mediated interactions. But I would be more interested in a more extensive discussion on how this is relevant for the actual understanding of the social deficits in the disorder.

We have rephrased this section and added several considerations, encompassing both our findings of equivalence in performance between groups and the possibility of encountering differences between groups at other levels of analysis. (lines 364-398, 412-423, 425-433 and 440-444). The authors acknowledge that, given the nature of the presented research, future work needs to be done to get a better understanding of the differences and similarities between CTRL and HFA across the multiple dimensions that embodied social interaction entails.

Reviewer 2 Report

The article by Zapata-Fonseca et al. explores the differences in sensitivity in social contingency in a group of high-functioning autism diagnosed adults, as compared to controls. The experiment of perceptual crossing is very well structured, simple to understand and scientifically well sound, although the sample is very small. Results are well presented. 

Some minor concerns should be addressed, according to my opinion:

- At page 3, line 101, the authors write “originally developed by [38]”. Please avoid this kind of error and report the name of the first author followed by the number of the reference list. Please, search overall the manuscript and correct the other similar mistakes.

- The methods are well presented, but, since the sample is small I think it will be possible to design a more precise table with the parameters of each subject analyzed for a better understanding of distribution frequencies, then reporting, as already done, the means. 

- At page 6, line 246 the phrase is truncated with an “and”. Please correct. 

- In the results section, for 3.2 and 3.3 paragraphs ti could be useful report tables with all the values, together with the figures, in order to easy the understanding. 

Author Response

Dear Reviewer, 

Many thanks for the observations. Here are our replies:

The article by Zapata-Fonseca et al. explores the differences in sensitivity in social contingency in a group of high-functioning autism diagnosed adults, as compared to controls. The experiment of perceptual crossing is very well structured, simple to understand and scientifically well sound, although the sample is very small. Results are well presented.

We thank the reviewer for the positive appraisal. Indeed, we have included a section on potential power issues in the discussion: “There are potential issues related to the latter, especially concerning the absence of significant interactions involving Group. It is possible that higher-order interactions remain elusive, that are related to how the HFA group may respond differently to the different objects than the Control group, or that their way of responding to said objects evolves differently over the course of the experiment. It should be noted however, that whereas all reported effect sizes were medium to large, none of the effect sizes involving Group reached the threshold for a small effect; still, the existence of higher-order effects cannot be excluded based on this small sample. What does seem clear is that, even if both groups could still differ to some extent in how they are sensitive to reactive versus non-reactive moving objects and how this extends over time, both groups are sensitive to an object’s reactivity to their own actions” (lines 420-429).

Some minor concerns should be addressed, according to my opinion:

- At page 3, line 101, the authors write “originally developed by [38]”. Please avoid this kind of error and report the name of the first author followed by the number of the reference list. Please, search overall the manuscript and correct the other similar mistakes.

This is now corrected

- The methods are well presented, but, since the sample is small I think it will be possible to design a more precise table with the parameters of each subject analyzed for a better understanding of distribution frequencies, then reporting, as already done, the means.

We are not entirely sure what the reviewer wants. Simply presenting the data of each participant and condition would involve creating three tables with 180 cells each, or 540 cells, which seems a bit much for such a short paper. As we have the data on OSF, we have opted to share the data in a link instead (see Figure 4 caption), which also encourages re-analysis and benefits exchange.

- At page 6, line 246 the phrase is truncated with an “and”. Please correct.

This is now corrected

- In the results section, for 3.2 and 3.3 paragraphs ti could be useful report tables with all the values, together with the figures, in order to easy the understanding.

As mentioned higher, this would lead to perhaps an overload of numbers, if the reviewer is referring to the data, and similarly for the complete results of the models (most of which are not only non-significant, but also of very (<small) effect size (which we now mention).
